# ONLINE IMPORTANCE SAMPLING FOR STOCHASTIC GRADIENT OPTIMIZATION

## ABSTRACT

Machine learning optimization often depends on stochastic gradient descent, where the precision of gradient estimation is vital for model performance. Gradients are calculated from mini-batches formed by uniformly selecting data samples from the training dataset. However, not all data samples contribute equally to gradient estimation. To address this, various importance sampling strategies have been developed to prioritize more significant samples. Despite these advancements, all current importance sampling methods encounter challenges related to computational efficiency and seamless integration into practical machine learning pipelines.

In this work, we propose a practical algorithm that efficiently computes data importance on-the-fly during training, eliminating the need for dataset preprocessing. We also introduce a novel metric based on the derivative of the loss w.r.t. the network output, designed for mini-batch importance sampling. Our metric prioritizes influential data points, thereby enhancing gradient estimation accuracy. We demonstrate the effectiveness of our approach across various applications. We first perform classification and regression tasks to demonstrate improvements in accuracy. Then, we show how our approach can also be used for *online* data pruning by identifying and discarding data samples that contribute minimally towards the training loss. This strategy yields significant reduction in training time with negligible to no loss in the accuracy of the model on unseen data.

## 1 INTRODUCTION

Stochastic gradient descent (SGD) combined with back-propagation has driven significant advances in optimization tasks. Its strength lies in its ability to optimize complex models by iteratively updating their parameters based on the gradient of the loss function. However, despite its widespread use, SGD has notable limitations. Convergence rates are influenced by several factors, with gradient noise being a key challenge that affects both robustness and convergence speed. Reducing this noise has been a focus of recent research (Alain et al., 2015; Faghri et al., 2020; Johnson & Zhang, 2013; Gower et al., 2020; Needell et al., 2014).

Various strategies have been proposed to mitigate gradient noise, including data diversification (Zhang et al., 2017; 2019), adaptive batch sizes, weighted sampling (Santiago et al., 2021), and importance sampling (Katharopoulos & Fleuret, 2018). These approaches aim to enhance gradient estimation and accelerate convergence in noisy optimization landscapes.

This work focuses on both importance sampling and data pruning as complementary techniques to improve training efficiency. Importance sampling involves constructing mini-batches through non-uniform data-point selection, i.e., picking certain data points with higher probability based on their expected contribution to the model's learning process. In parallel, data pruning seeks to identify and eliminate data points that contribute minimally to training, reducing computational load. This is especially beneficial in large-scale learning tasks, where reducing data complexity can significantly improve both time and resource efficiency. By jointly leveraging these two techniques, we aim to both improve the accuracy of gradient estimates and streamline the training process by focusing computation resources on valuable data.

In this paper, we introduce a novel metric that quantifies the contribution of each data sample to the model's learning process, to guide both importance sampling and data pruning decisions. Our

approach leverages information from the network's output to strategically allocate computational resources to the most impactful data points. This results in substantial improvements in convergence across a variety of tasks, while sustaining minimal computational overhead compared to state-of-the-art methods that share similar goals (Katharopoulos & Fleuret, 2018; Santiago et al., 2021).

In summary, our contributions can be distilled into the following key points:

- We propose an adaptive metric for importance sampling improving gradient accuracy.
- We introduce an efficient online sampling algorithm that incorporates our metric.
- We demonstrate the effectiveness of our approach through evaluations on classification and regression problems.
- We further demonstrate the ability of our algorithm to perform online data pruning. Our approach allows using any importance function for data pruning and does not require any pre-processing of the data.

## 2  RELATED WORK

Gradient estimation is a cornerstone in machine learning, underpinning the optimization of models. In practical scenarios, computing the exact gradient is infeasible due to the sheer volume of data, leading to the reliance on mini-batch approximations. Improving these approximations to obtain faster and more accurate estimates remains a challenge. The ultimate goal is to accelerate gradient descent by using more accurate gradient estimates.

**Importance sampling.**  Importance sampling serves as a mechanism for error reduction in mini-batch gradient estimation. Each data point is assigned a probability to be selected in each mini-batch, making some data more likely to be chosen than others. Bordes et al. (2005) developed an online algorithm (LASVM) which uses importance sampling to train kernelized support vector machines. Several studies have shown that importance sampling proportional to the gradient norm is the optimal sampling strategy (Zhao & Zhang, 2015; Needell et al., 2014; Wang et al., 2017; Alain et al., 2015). Hanchi et al. (2022) recently proposed deriving an importance sampling metric from the gradient norm of each data point, demonstrating favorable convergence properties and provable improvements under certain convexity conditions.

Estimating the gradient for each data point can be computationally intensive. Thus, the search for more efficient sampling strategies has led to the exploration of efficient approximations of the gradient norm. Methods proposed by Loshchilov & Hutter (2015) rank data based on their loss and derive an importance sampling strategy assigning higher importance to data with higher loss. Katharopoulos & Fleuret (2017) proposed importance sampling the loss function. Additionally, Dong et al. (2021) proposed a resampling-based algorithm to reduce the number of backpropagation computations, selecting a subset of data based on the loss. Similarly, Zhang et al. (2023) proposed resampling based on multiple heuristics to reduce the number of backward propagations and focus on more influential data. Katharopoulos & Fleuret (2018) introduced an upper bound to the gradient norm that can be used as an importance function, suggesting re-sampling data based on importance computed at the last layer. These resampling methods reduce unnecessary backward propagations but still require forward computation for each data point.

**Data weighting.**  An alternative to importance sampling is to adjust the contribution of uniformly selected data points by a weighting factor. To compute weights within a mini-batch, Santiago et al. (2021) proposed a method maximizing the mini-batch's effective gradient. This allocation of weights aims to align data contributions with the optimization objective, expediting convergence at the cost of potential bias.

**Data pruning.**  Data pruning reduces the computational load of training by removing minimally useful data. Early work by Har-Peled & Kushal (2005) proposed using a smaller, representative dataset for k-means clustering. This concept has expanded to other machine learning tasks, where not all data points contribute equally to learning. Toneva et al. (2019) found that some data points, once correctly classified, remain so, suggesting they can be pruned without affecting performance.

Coleman et al. (2020) introduced a proxy network to guide pruning by selecting relevant data points based on predictions. Paul et al. (2021) further refined this strategy by using early training information to identify important data points, allowing training on smaller data subsets with small performance loss. These methods show that focusing on the most informative samples can enhance training efficiency. Yang et al. (2023) proposed to select a subset of the dataset and propose a discrete optimization method using influence functions to determine which data points to retain and which to prune from the training dataset. Unfortunately, their overall preprocessing can take hours and does not scale well to large datasets. In contrast, our importance sampling algorithm can be used for *online* data pruning without any preprocessing.

## 3 BACKGROUND

In machine learning, the goal is to find the optimal set of parameters $\theta$ for a model function $m(x, \theta)$, with $x$ a data sample (and $y$ its supervision label), that minimize a loss function $\mathcal{L}$ over a dataset $\Omega$. The optimization is typically expressed as

$$\theta^* = \underset{\theta}{\operatorname{argmin}} \, L_\theta, \quad \text{where} \quad L_\theta = \frac{1}{|\Omega|} \int_\Omega \mathcal{L}(m(x, \theta), y) \mathrm{d}(x, y) = \mathrm{E}\left[\frac{\mathcal{L}(m(x, \theta), y)}{p(x, y)}\right]. \quad (1)$$

The total loss $L_\theta$ can be interpreted in two ways. The analytical interpretation views it as the integral of the loss $\mathcal{L}$ over a data space $\Omega$, normalized by the space's volume. In machine learning, the data space typically represents the (discrete) training dataset and the normalization is its size. The second, statistical interpretation defines $L_\theta$ as the expected value of the loss $\mathcal{L}$ for a randomly selected data point, divided by the probability of selecting it. The two approaches are equivalent.

In practice, the minimization of the total loss $L_\theta$ is tackled via iterative gradient descent. At each iteration $t$, its gradient $\nabla L_{\theta_t}$ with respect to the current model parameters $\theta_t$ is computed, and those parameters are updated as

$$\theta_{t+1} = \theta_t - \lambda \nabla L_{\theta_t}, \quad (2)$$

where $\lambda > 0$ is the learning rate. The procedure is repeated until convergence.

### 3.1 MONTE CARLO GRADIENT ESTIMATOR

**Gradient estimator.** The parameter update in Eq. (2) involves evaluating the total-loss gradient $\nabla L_{\theta_t}$. This requires processing the entire dataset $\Omega$ at each of potentially many (thousands of) steps, making the optimization computationally infeasible. In practice one has to resort to mini-batch gradient descent which estimates the gradient from a small set $\{x_i\}_{i=1}^B \subset \Omega$ of randomly chosen data points in a Monte Carlo fashion:

$$\nabla L_\theta \approx \frac{1}{B} \sum_{i=1}^B \frac{\nabla \mathcal{L}(m(x_i, \theta), y_i)}{p(x_i, y_i)} = \langle \nabla L_\theta \rangle, \quad \text{with} \quad x_i \propto p(x_i). \quad (3)$$

Here, $\nabla \mathcal{L}(m(x_i, \theta), y_i)$ is the gradient (w.r.t. $\theta$) of the loss function for sample $x_i$ selected following a probability density function (pdf) $p$ (or probability mass function in case of a discrete dataset). Any distribution $p$ ensuring that $p(x) = 0 \Rightarrow \nabla \mathcal{L}(m(x_i, \theta), y_i) = 0$ yields an unbiased gradient estimator, i.e., $\mathrm{E}[\langle \nabla L_\theta \rangle] = \nabla L_\theta$. Mini-batch gradient descent uses $\langle \nabla L_\theta \rangle$ in place of the true gradient $\nabla L_\theta$ in Eq. (2) to update the model parameters at every optimization iteration. The batch size $B$ is typically much smaller than the dataset, enabling practical optimization.

**Theoretical convergence analysis.** Mini-batch gradient descent is affected by Monte Carlo noise due to the stochastic gradient estimation in Eq. (3). This noise arises from the varying contributions of different samples $x_i$ to the estimate and can cause the parameter optimization trajectory to be erratic, slowing down convergence. In certain conditions, it is possible to express the convergence rate of such methods. Gower et al. (2019) demonstrated that for an $L$-smooth and $\mu$-convex function, the convergence rate of mini-batch gradient descent with constant learning rate is

$$\mathrm{E}\left[\|\theta_t - \theta^*\|^2\right] \leq (1 - \lambda\mu)^t \|\theta_0 - \theta^*\|^2 + \frac{2\lambda\sigma^2}{\mu}, \quad (4)$$

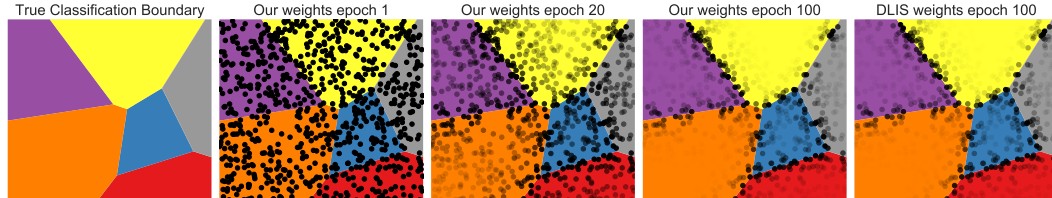

Figure 1: Visualization of the importance sampling at 3 different epoch and the underlying classification task. For each presented epoch, 800 data-point are presented with a transparency proportional to their weight according to our method.

with $\sigma^2 = \mathrm{E}\left[\|\langle \nabla L_{\theta^*}\rangle\|^2\right] - \overbrace{\mathrm{E}\left[\|\langle \nabla L_{\theta^*}\rangle\|\right]^2}^{=0}$. The expected value of the gradient norm is zero for the optimal set of parameters $\theta^*$, as the solution of the gradient descent is reached when the gradient converges to zero. This equation underscores the significance of minimizing variance in gradient estimation to enhance the convergence rate of gradient descent methods. While not universally applicable, it provides valuable insights into expected behavior when reducing estimation errors. Hence, refining gradient estimates is crucial for optimizing various learning algorithms, facilitating more efficient convergence towards optimal solutions. Our experimental evaluation comparing different methods in Section 4.2 further supports this notion.

## 4 IMPORTANCE FUNCTION

### 4.1 GRADIENT NORM BOUND

The gradient $L_2$ norm has been shown to be an optimal choice of importance sampling (Zhao & Zhang, 2015; Needell et al., 2014; Wang et al., 2017; Alain et al., 2015) as it minimizes the first term of the gradient variance, thereby bounding the convergence of Eq. (4). However, calculating it requires costly full backpropagation for every data point, which is what we want to avoid in the first place. Instead, we compute an upper bound of the gradient norm using the output nodes of the network: $q(x) = \left\|\frac{\partial \mathcal{L}(x)}{\partial m(x,\theta)}\right\|$. This upper bound of the gradient norm is derived from the chain rule and the Cauchy–Schwarz inequality:

$$\left\|\frac{\partial \mathcal{L}(x_i)}{\partial \theta}\right\| = \left\|\frac{\partial \mathcal{L}(x)}{\partial m(x,\theta)} \cdot \frac{\partial m(x,\theta)}{\partial \theta}\right\| \leq \left\|\frac{\partial \mathcal{L}(x)}{\partial m(x,\theta)}\right\| \cdot \left\|\frac{\partial m(x,\theta)}{\partial \theta}\right\| \leq \underbrace{\left\|\frac{\partial \mathcal{L}(x)}{\partial m(x,\theta)}\right\|}_{q(x)} \cdot C, \quad (5)$$

where $C$ is the Lipschitz constant of the parameters gradient. That is, our importance function is a bound of the gradient magnitude based on the output-layer gradient norm. For specific shapes of the output layer, it is possible to derive a closed form expression. Below we show such derivation for classification networks based on the cross-entropy loss.

**Cross-entropy loss gradient.** Cross entropy is the standard loss function in classification tasks. It quantifies the dissimilarity between predicted probability distributions and actual class labels. Specifically, for a binary classification task, cross entropy is defined as:

$$\mathcal{L}(m(x_i,\theta)) = -\sum_{j=1}^{J} y_j \log(s_j) \ \text{ where } s_j = \frac{\exp(m(x_i,\theta)_j)}{\sum_{k=0}^{J} \exp(m(x_i,\theta)_k)} \quad (6)$$

where $m(x_i,\theta)$ is an output layer, $x_i$ is the input data and $J$ means the number of classes. It is possible to express the derivative of the loss $\mathcal{L}$ with respect to the network output $m(x_i,\theta)_j$ in a close form.

$$\frac{\partial \mathcal{L}}{\partial m(x_i,\theta)_j} = s_j - y_j \quad (7)$$

This equation can be directly computed from the network output without any graph backpropagation. This make the computation of our importance function extremely cheap for classification tasks. Proof of the derivation can be found in the Appendix A.

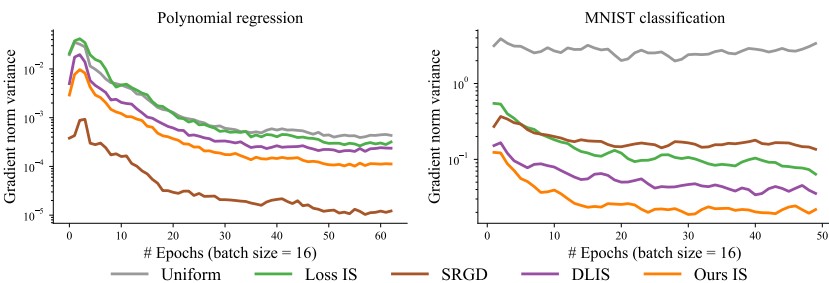

Figure 2: Evolution of gradient variance for variance importance sampling strategies on polynomial regression and MNIST classification task. In both case the optimization is done on a 3 fully-connected layer network. Variance estimation is made of each method on the same network at each epoch. The variance is computed using a mini-batch of size 16. Computation time for each metric can be found in Appendix D Table 2

Importance sampling in classification emphasizes gradients along classifiction boundaries, where parameter modifications have the greatest impact. Figure 1 illustrates this concept, showing iterative refinement of the sampling distribution to focus on boundary decisions in comparison to data within classes. The rightmost column illustrates the sampling distribution of the DLIS method of Katharopoulos & Fleuret (2018) at epoch 100. Both methods iteratively increase the importance of the sampling around the boundary decision compare to data inside the classes.

Our approach differs from that of Katharopoulos & Fleuret in that we compute the gradient norm with respect to the network's output logits. This approach often allows gradient computation without requiring back-propagation or graph computations, streamlining optimization.

## 4.2 CONVERGENCE ANALYSIS

Building on the theoretical bound defined in Eq. (4), we proceed to examine the effects of various importance sampling methods on the gradient variance. Such variance influences the convergence of an optimization procedure. This equation relies on the ideal model parameters $\theta^*$, but they cannot be practically calculated. Rather, we measure the gradient variance during training using a suboptimal parameter set.

Figure 2 displays the evolution of gradient variance using different strategies for polynomial regression and MNIST classification, both using a three-layer fully connected network. Each method is evaluated on the same network, trained using uniform sampling. This allows for a variances comparison of the gradient norm. We analyze five techniques: Uniform sampling, Loss-based importance sampling, SRGD (Hanchi et al., 2022), DLIS (Katharopoulos & Fleuret, 2018), and our method. SRGD is an importance sampling technique using a conditioned minimization of gradient variance using memory of the gradient magnitude. This method has shown robust theoretical convergence properties in strongly convex scenarios. This variance reduction is visible on the polynomial regression task where it result in lower variance than other methods. However, for more complex tasks such as MNIST classification, SRGD underperforms all methods, suggesting scalability limitations to non-convex and complex problems. In contrast, our method consistently yields lower variance than Loss-based importance sampling and DLIS (Katharopoulos & Fleuret, 2018). These findings elucidate the results in Section 7.

In addition, we provide evaluation times for each metric for both the polynomial regression and MNIST classification tasks in Appendix D Table 2. Clearly, SRGD (Hanchi et al., 2022) demands more computational resources, even for small-scale networks comprising only three layers. This increased demand stems from its dependence on calculating the gradient norm for each individual data point. Both our metric, which employs automatic differentiation, and DLIS (Katharopoulos & Fleuret, 2018), incur comparable computational costs due to their reliance on derivatives from the final layers. Nonetheless, our approach proves to be the most efficient when analytical evaluations are feasible. Such differences in computational efficiency are likely to significantly influence the outcomes of comparisons made under equal-time conditions in later sections.

Zhao & Zhang (2015) have shown that importance weights w.r.t. the gradient norm gives the optimal sampling distribution. On the right inline figure, we show the difference between various weighting strategies and the gradient norm w.r.t. all parameters. In this experiment, all sampling weights are computed using the same network on an MNIST optimization task. Our proposed sampling strategies, based on the loss gradient are the closest approximation to the gradient norm.

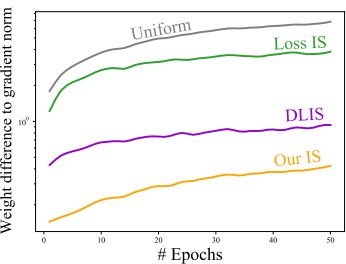

## 5 ONLINE IMPORTANCE SAMPLING ALGORITHM

We propose an algorithm to efficiently perform importance sampling for mini-batch gradient descent, outlined in Algorithm 1. Similarly to Loshchilov & Hutter (2015) and Schaul et al. (2015), it is designed to use an importance function that relies on readily available quantities for each data point, introducing only negligible memory and computational overhead over classical uniform mini-batching.

---

**Algorithm 1** Mini-batch importance sampling for SGD.

1: $\theta \leftarrow$ random parameter initialization
2: $B \leftarrow$ mini-batch size, $N = |\Omega|$              ← Dataset size
3: $q, \theta \leftarrow \text{Initialize}(\Omega, \theta, B)$              ← Algorithm 3
4: **until** convergence **do**              ← Loop over epochs
5:    **for** $t \leftarrow 1$ **to** $N/B$ **do**           ← Loop over mini-batches
6:      $p \leftarrow q/\text{sum}(q)$           ← Normalize importance to pdf
7:      $x, y \leftarrow B$ data samples $\{x_i, y_i\}_{i=1}^{B} \propto p$
8:      $\mathcal{L}(x) \leftarrow \mathcal{L}(m(x, \theta), y)$
9:      $\nabla \mathcal{L}(x) \leftarrow \text{Backpropagate}(\mathcal{L}(x))$
10:      $\langle \nabla L_\theta \rangle \leftarrow (\nabla \mathcal{L}(x) \cdot (1/p(x))^T)/B$      ← Eq. (3)
11:      $\theta \leftarrow \theta - \eta \langle \nabla L_\theta \rangle$          ← SGD step
12:      $q(x) \leftarrow \alpha \cdot q(x) + (1 - \alpha) \cdot \left\| \frac{\partial \mathcal{L}(x)}{\partial m(x,\theta)} \right\|$    ← Accumulate importance
13:    $q \leftarrow q + \epsilon$
14: **return** $\theta$

---

We maintain a set of persistent *un-normalized importance* scalars $q = q_{i=1}^{|\Omega|}$, continually updated during optimization. Initially, we process all data points once in the first epoch to determine their initial importance (line 3). Subsequently, at each mini-batch optimization step $t$, we normalize the importance values to obtain the probability density function (PDF) $p$ (line 6), and use it to sample $B$ data points with replacement (line 7). We then evaluate the loss for each selected data sample (line 8) and backpropagate to compute the corresponding loss gradient (line 9). Finally, we update the network parameters using the estimated gradient (line 11). Additionally, we compute the sample importance for each data sample from the mini-batch and update the persistent importance $q$ (line 12). Various importance heuristics such as the gradient norm (Zhao & Zhang, 2015; Needell et al., 2014; Wang et al., 2017; Alain et al., 2015), the loss (Loshchilov & Hutter, 2015; Katharopoulos & Fleuret, 2017; Dong et al., 2021) or more advanced importance (Katharopoulos & Fleuret, 2018) can be implemented to replace our sampling metric in this line. To enhance efficiency, our algorithm reuses the forward pass computations made during line 8 to compute importance, updating $q$ only for the current mini-batch samples. The weighting parameter $\alpha$ ensures weight stability as discussed in Eq. (8).

At the end of each epoch (line 14), we add a small value to the un-normalized weights of all data to ensure that every data point will be eventually evaluated, even if its importance is deemed low by the importance metric.

It is importance to note that the initialization epoch is done without importance sampling to initialize each sample importance. This does not create overhead as it is equivalent to a classical epoch running over all data samples. While similar schemes have been proposed in the past, they often rely on a multitude of hyperparameters, making their practical implementation challenging. This has led to the development of alternative methods like re-sampling (Katharopoulos & Fleuret, 2018;

Dong et al., 2021; Zhang et al., 2023). Our proposed sampling strategy has only a few hyperparameters. Tracking importance across batches and epochs minimizes the computational overhead, further enhancing the efficiency and practicality of the approach.

# 6 ONLINE DATA PRUNING

Data pruning is a technique aimed at reducing the size of the dataset to accelerate training. The acceleration can be attributed to two main factors. The first, and most practical, relates to the execution speed of training neural networks. When working with large datasets, especially those with a relatively large memory footprint, it is often infeasible to store all data directly in GPU memory. This necessitates frequent data loading from slower storage mediums, which can become a bottleneck and significantly slow down training. By reducing the dataset size, less data needs to be loaded during each training iteration, leading to faster execution, even if the theoretical properties of the training process remain unchanged. The second factor contributing to faster training is theoretical. If the pruned data points have a low gradient norm, removing them increases the expected gradient norm of the remaining data points. This, in turn, leads to larger effective steps in the optimization process, thus accelerating convergence.

Given these two benefits, we propose a data pruning strategy guided by our novel importance metric, which serves as an estimate of the gradient norm for each data point. Unlike most previous works that rely on precomputed metrics or early-stage proxies, our metric is adaptive throughout training and does not require any precomputation. This allows us to dynamically prune the dataset based on current information about the importance of each data point.

---

**Algorithm 2** Subroutine for data pruning

1: **function** ONLINEDATAPRUNING($\Omega, q, K$)
2: $\quad \epsilon \leftarrow \frac{1}{K|\Omega|} \sum_{x \in \Omega} q(x)$          $\leftarrow$ Compute pruning threshold
3: $\quad \Omega' \leftarrow \Omega_{\{q(x) > \epsilon | \forall x \in \Omega\}}$      $\leftarrow$ Filter dataset to keep high importance data
4: $\quad$ **return** $\Omega'$

---

Our approach involves an online pruning process that operates as follows: After a certain number of epochs, we ensure that the importance metric has been calculated for all data points in the training set. At this point, we identify and remove a portion of the data with importance metrics significantly lower than the average. Specifically, each data point's importance is compared to the average importance across the dataset. If a data point's importance falls below a threshold relative to the average, it is pruned from the training set. This ensures that only data points with low expected gradient norms are removed, while important data remains. Algorithm 2 depicts the pruning subroutine, which processes the dataset $\Omega$, the importance score for each data point $q$, and a reduction factor $K$. A higher reduction factor results in retaining more data points, thus fewer data are pruned.

This process is flexible and can adapt to the distribution of importance values in the dataset. If the dataset has a wide distribution of importance, with only a few data points contributing significantly to the optimization, a large portion of the dataset can be pruned. Conversely, if all data points exhibit relatively high importance, few or no data points will be removed. Furthermore, since our importance metric is adaptive, this pruning process can be applied multiple times throughout training. By continually updating the importance metric and pruning low-importance data points, we maintain an efficient training set that accelerates the learning process without compromising model performance.

# 7 EXPERIMENTS

In this section, we delve into the experimental outcomes of our proposed algorithm and sampling strategy. Our evaluations encompass diverse classification and regression tasks. We benchmarked our approach against those of Katharopoulos & Fleuret (2018) and Santiago et al. (2021), considering various variations in comparison. Distinctions in our comparisons lie in assessing performance at equal steps/epochs and equal time intervals. The results presented here demonstrate the loss and classification error, computed on test data that remained unseen during the training process.

Table 1: We compare the impact of our importance sampling algorithm with or without data pruning on classification tasks. We compare on three different datasets: Point cloud, CIFAR-100 and Tiny-ImageNet. Our approach consistently outperform on majority of datasets (see Table 3 for more comparisons). Bold numbers represents the best scores, underlined ones represent the second best.

| Method | Point cloud | | | | CIFAR-100 | | | | Tiny-ImageNet | | | |
|---|---|---|---|---|---|---|---|---|---|---|---|---|
| | Equal step | | Equal time | | Equal step | | Equal time | | Equal step | | Equal time | |
| | Loss (↓) | Accuracy (↑) | Loss | Accuracy | Loss | Accuracy | Loss | Accuracy | Loss | Accuracy | Loss | Accuracy |
| Uniform | 0.00505 | 82.3 | 0.00505 | 82.2 | 0.012 | 72.6 | 0.012 | 73.8 | 0.02176 | 47.4 | 0.02176 | 47.4 |
| Loss IS | 0.00495 | 82.6 | 0.00496 | 82.5 | 0.023 | 40.2 | 0.026 | 32.8 | 0.02237 | 45.2 | 0.02238 | 45.2 |
| DLIS | 0.00595 | 81.9 | 0.00603 | 81.8 | 0.015 | 62.0 | 0.015 | 60.8 | 0.03433 | 26.3 | 0.03433 | 26.3 |
| DLIS weights w/ Our algorithm | 0.00481 | 82.6 | 0.00485 | 82.5 | 0.021 | 43.6 | 0.029 | 26.3 | 0.03454 | 26.1 | 0.02778 | 35.0 |
| LOW | 0.00572 | 82.6 | 0.01173 | 74.9 | 0.011 | **74.6** | 0.011 | 74.2 | 0.02344 | 43.5 | 0.02344 | 43.5 |
| Our IS | 0.00480 | 82.9 | 0.00480 | 82.9 | 0.011 | 74.3 | 0.012 | 74.3 | **0.02127** | 48.1 | **0.02123** | **48.5** |
| Our IS + Data pruning | **0.00478** | **83.2** | **0.00478** | **83.1** | **0.011** | 74.3 | **0.011** | **74.3** | 0.02193 | 47.1 | 0.02193 | 47.1 |

## 7.1 IMPLEMENTATION DETAILS

We implement our method and all baselines in a single PyTorch framework. Experiments run on a workstation with an NVIDIA Tesla A40 graphics card. The baselines include uniform sampling, DLIS (Katharopoulos & Fleuret, 2018) and LOW (Santiago et al., 2021). Uniform means that we sample every data point from a uniform distribution. DLIS importance samples the data mainly depending on the norm of the gradient on the last output layer. We use functorch (Horace He, 2021) to accelerate this gradient computation. LOW is based on adaptive weighting that maximizes the effective gradient of the mini-batch using the solver from Vandenberghe (2010).

We evaluated our method on a range of tasks, including image classification with MNIST, CIFAR-10/100 (Krizhevsky et al., 2009), Tiny-ImageNet (Le & Yang, 2015), and Oxford Flower-102 (Nilsback & Zisserman, 2008), as well as Point cloud classification (Qi et al., 2017) and regression tasks (Sitzmann et al., 2020). Full details on the datasets used, along with optimization parameters such as learning rate, optimizer scheduler and the data pruning ratio and frequency are provided in Appendix C. In all results involving pruning, the number of steps per epoch remains consistent with the non-pruned experiments. This ensures a fair comparison at equal steps, meaning that with pruning, certain data points are seen multiple times within each epoch to match the total step count.

**Weight stability.** Updating the persistent per-sample importance $q$ directly sometime leads to a sudden decrease of accuracy during training. To make the training process more stable, we update $q$ by linearly interpolating the importance at the previous and current steps:

$$q(x) = \alpha \cdot q_{prev}(x) + (1 - \alpha) \cdot q(x) \tag{8}$$

where $\alpha$ is a constant for all data samples. In practice, we use $\alpha \in \{0.0, 0.1, 0.2, 0.3\}$ as it gives the best trade-off between importance update and stability. This can be seen as a momentum evolution of the per-sample importance to avoid high variation. Utilizing an exponential moving average to update the importance metric prevents the incorporation of outlier values. This is particularly beneficial in noisy setups, like situations with a high number of class or a low total number of data. Details on the chosen $\alpha$ values can be found in Appendix C.

## 7.2 RESULTS

In Table 1, we compare Uniform sampling, Loss-based importance sampling, the method from Katharopoulos & Fleuret (2018) and their weights in our algorithm, the approach from Santiago et al. (2021), and our method with both importance sampling and data pruning. The table reports the cross-entropy loss and classification accuracy for three tasks: point cloud classification, CIFAR-100, and Tiny-ImageNet. Results are shown for both an equal number of steps and equal runtime. The best results are highlighted in bold, with the second-best underlined. Across all three tasks, our method consistently achieves the best performance in both scenarios. Even in cases where importance sampling offers minimal improvement, our approach proves more robust than DLIS and LOW, avoiding significant underperformance in challenging situations. In the Tiny-ImageNet experiment, although data pruning results in a slight drop in accuracy, the outcome aligns with the

| Method | Accuracy | Remaining data (At end of opt.) | Training time(s) (Pruning time) |
|---|---|---|---|
| Uniform - - | 96.4% | 100% | 554 (0) |
| Random pruning - - | 96.3% | 60% | 131 (3) |
| Random pruning - - | 96.2% | 43% | 121 (2) |
| Random pruning - - | 96.1% | 35% | 114 (1) |
| Yang et al. (2023) - - | 96.4% | 60% | 132 (2388) |
| Yang et al. (2023) - - | 96.3% | 43% | 128 (4825) |
| Yang et al. (2023) - - | 96.2% | 35% | 121 (9351) |
| **Ours** (K=8) —— | 97.9% | 62% (33%) | 215 (3) |
| **Ours** (K=4) —— | 98.1% | 45% (15%) | 214 (2) |
| **Ours** (K=2) —— | 98.1% | 34% (6%) | 208 (2) |
| **Ours** (K=1) —— | 92.6% | 27%(0.6%) | 203 (2) |

Figure 3: Evaluation of the impact of the amount of data pruned during training on a MNIST classification task. The left panel shows the evolution of the pruned data over time, while the right panel presents the final accuracy, the average training set size during training and remaining data at the end of training, the total training time, and the computation time of pruning. The figure compares a uniform sampling without data pruning, random pruning with $60\%$, $43\%$, and $35\%$ of data pruned, the method of Yang et al. (2023) at the same pruning rates, and our approach using a dynamic reduction factor $K$. Results indicate that pruning more data accelerates execution. Our online pruning method offers greater adaptability during training while maintaining high accuracy and minimal difference between training time and total execution time.

observations from Yang et al. (2023), where pruning can leads to a small reduction in generalization. Additional results on other datasets can be found in Table 3.

Figure 4 illustrates the results of a regression task on an image using a SIREN network to learns the mapping between 2D pixel coordinates and the corresponding RGB color. The left panel shows the loss evolution for all methods, while the right panel presents the error maps at final steps, along with a zoomed-in region for Uniform sampling, DLIS, our method with importance sampling, and our method combining importance sampling and data pruning. Our method, which incorporates importance sampling and data pruning, provides the best loss reduction performance. The error map reveals fewer errors, with less yellow tones and finer details in the zoomed region. This method effectively reduces error in high-frequency regions by compensating in smoother regions such as the background, leading to a more balanced error distribution across the image. In comparison, DLIS produces similar results to our importance sampling when its weights are used with our algorithm, but its full method is significantly outperformed by Uniform sampling. This is evident in the error map and the zoomed-in area, which display more blurriness in DLIS results.

Figure 3 presents an ablation study of our online pruning strategy on the MNIST classification task, comparing random pruning, the method of Yang et al. (2023), and our adaptive approach. The left side shows the evolution of the data used during training across epochs, while the right side highlights final accuracy, the average data used per epoch and the final remaining data, the training time and time used to compute the pruning. Our adaptive method starts with the full dataset and prunes data every 20 epochs, following Algorithm 2. As training progresses, the amount of pruned data decreases, since many data points begin to contribute redundant information. By the end, only a reduced subset remains. In contrast, Yang et al. (2023) prunes in one step at the start using a pre-trained model, leading to faster training but lower quality results, not outperforming uniform sampling but providing generalization properties. Our approach adaptively removes data that no longer adds value to the learning process. While aggressive pruning (e.g., $K = 1$) risks overfitting and reduced accuracy, more moderate pruning speeds up training without sacrificing quality. The results show that when minimal or no pruning is applied, the advantages diminish, reverting to a reliance on importance sampling alone. Overall, our adaptive pruning efficiently reduces dataset size while preserving crucial data throughout training.

**Additional experiments.** Further comparisons, similar to those in Table 1, across various datasets are provided in Appendix D. We also present convergence curves at equal steps and equal time intervals for Pointnet, CIFAR-10 (ViT (Dosovitskiy et al., 2021)) and Tiny-ImageNet, demonstrating

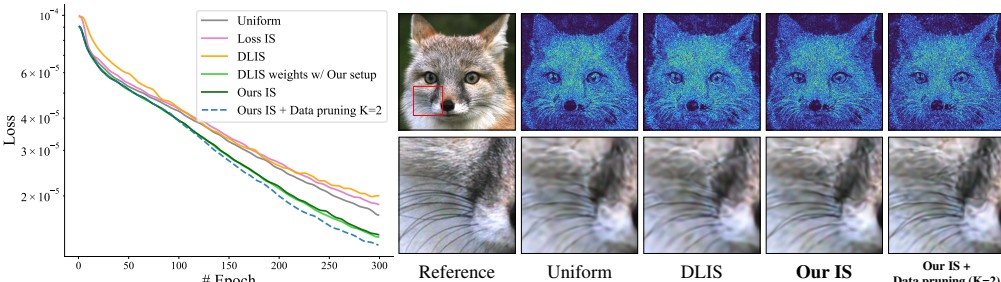

Figure 4: Comparison at equal step for image 2D regression. The left side shows the convergence plot while the right display the absolute error of the regression and a close-up view. Our method using data pruning achieves the lower error on this problem while pruning 45% of the data during training. Our method using only importance sampling and DLIS with our algorithm perform similarly, but DLIS with their full method perform worse than default optimization. In the images it is visible that our method with pruning recovers the finest details of the fur and whiskers.

the consistent improvements of our method throughout the optimization process. These additional experiments reinforce the effectiveness of our approach and in particular benefit from a low computation method at equal time.

**Discussion.** Both our and DLIS importance metrics are highly correlated, but ours is simpler and more efficient to evaluate. Even with a slightly better importance sampling metric, most of the improvement come from the memory-based algorithm instead of a resampling one. The resulting algorithm gives better performance at the same time and has more stable convergence. Our online data pruning method is controlled by a pruning factor $K$, which dictates how much data is removed at each step. While we kept $K$ constant in our experiments, it could be adjusted to prevent overfitting. Pruned data could also be reintroduced later to check for overfitting by observing if its importance increases after removal. This could help detect reduced generalization without shrinking the initial dataset, though we leave this for future work.

**Limitations.** As the algorithm rely on past information to drive a non-uniform sampling of data, it requires seeing the same data multiple times. This creates a bottleneck for architectures that rely on progressive data streaming. More research is needed to design importance sampling algorithms for data streaming architectures, which is a promising future direction. Non-uniform data sampling can also create slower runtime execution. The samples selected in a mini-batch are not laid out contiguously in memory leading to a slower loading. We believe a careful implementation can mitigate this issue.

## 8 CONCLUSION

In conclusion, our work introduces an efficient sampling strategy for machine learning optimization, that can be use for importance sampling and data pruning. This strategy, which relies on the gradient of the loss and has minimal computational overhead, was tested across various classification as well as regression tasks with promising results. Our work demonstrates that by paying more attention to samples with critical training information, we can speed up convergence without adding complexity. We hope our findings will encourage further research into simpler and more effective sampling strategies for machine learning.

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

## A  DERIVATIVE OF CROSS-ENTROPY LOSS

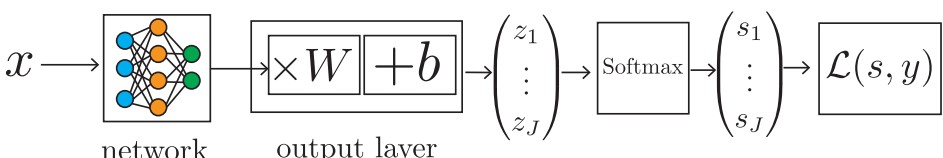

Machine learning frameworks take data $x$ as input, performs matrix multiplication with weights and biases added. The output layer is then fed to the softmax function to obtain values $s$ that are fed to the loss function. $y$ represents the target values. We focus on the categorical cross-entropy loss function for the classification problem (with $J$ categories) given by:

$$\mathcal{L}_{\text{cross-ent}} = -\sum_i y_i \log s_i \ \text{ where } s_i = \frac{\exp(m(x_i,\theta)_l)}{\sum_l^J \exp(m(x_i,\theta)_l)} \tag{9}$$

For backpropagation, we need to calculate the derivative of the $\log s$ term wrt the weighted input $z$ of the output layer. We can easily derive the derivative of the loss from first principles as shown below:

$$\frac{\partial \mathcal{L}_{\text{cross-ent}}}{\partial m(x_i,\theta)_j} = -\frac{\partial}{\partial m(x_i,\theta)_j}\left(\sum_i^J y_i \log s_i\right) = -\sum_i^J y_i \frac{\partial}{\partial m(x_i,\theta)_j}\log s_i = -\sum_i^J \frac{y_i}{s_i}\frac{\partial s_i}{\partial m(x_i,\theta)_j} \tag{10}$$

$$= -\sum_i^J \frac{y_i}{s_i} s_i \cdot (\mathbf{1}\{i == j\} - s_j), \text{ can be easily derived from first principles,} \tag{11}$$

$$= \sum_i^J y_i \cdot s_j - \sum_i^J y_i \cdot (\mathbf{1}\{i == j\}) = s_j \sum_i^J y_i - y_j = s_j - y_j \tag{12}$$

The partial derivative of the cross-entropy loss function wrt output layer parameters has the form:

$$\frac{\partial \mathcal{L}_{\text{cross-ent}}}{\partial m(x_i,\theta)_j} = s_j - y_j \tag{13}$$

For classification tasks, we directly use this analytic form of the derivative and compute it's norm as weights for importance sampling.

## B  ALGORITHM DETAILS

Algorithm 3 provide detail on the initialization subroutine applying a first epoch of training without importance sampling to initialize the persistent importance vector $q$.

## C  DATASET AND TRAINING DETAILS

In this section we provide details of the datasets and training. We train all models with 3 independent runs and report the average loss and accuracy as shown in Table 3.

---

**Algorithm 3** Subroutine for initialization for Algorithm 1

---

1: **function** INITIALIZATION($\Omega,\theta,B,q$)                                                    $\leftarrow$ Initialize $q$ in a classical SGD loop
2:    **for** $t \leftarrow 1$ **to** $|\Omega|/B$ **do**
3:       $x, y \leftarrow \{x_i, y_i\}_{i=(t-1)\cdot B+1}^{t\cdot B+1}$                   $\leftarrow$ See all samples in the first epoch
4:       $\mathcal{L}(x) \leftarrow \mathcal{L}(m(x,\theta), y)$
5:       $\nabla\mathcal{L}(x) \leftarrow \text{Backpropagate}(\mathcal{L}(x))$
6:       $\langle\nabla L_\theta\rangle(x) \leftarrow \nabla\mathcal{L}(x)/B$                                      $\leftarrow$ Eq. (3)
7:       $\theta \leftarrow \theta - \eta\langle\nabla L_\theta\rangle(x)$                                         $\leftarrow$ Eq. (2)
8:       $q(x) \leftarrow \left\|\frac{\partial\mathcal{L}(x)}{\partial m(x,\theta)}\right\|$         $\leftarrow$ Initialize per sample importance
9:    **return** $q,\theta$

---

**MNIST.** The MNIST database contains 60,000 training images and 10,000 testing images. We train a 3-layer fully-connected network (MLP) for image classification over 50 epochs with an Adam optimizer (Kingma & Ba, 2014).

**CIFAR-10 and CIFAR-100.** CIFAR-10 (Krizhevsky et al., 2009) contains 60,000 32x32 color images from 10 different object classes, with 6,000 images per class. CIFAR-100 (Krizhevsky et al., 2009) has 100 classes containing 600 images each, with 500 training images and 100 testing images per class. For both datasets, we use the ResNet-18 network architecture (He et al., 2016). We use the SGD optimizer with momentum 0.9, initial leaning rate 0.003 (CIFAR-10) and 0.007 (CIFAR-100), and batch size 128. We reduced the initial learning rate following an exponential scheduling with factor 0.987 over a total of 120 epochs for CIFAR-10 and 200 epochs for CIFAR-100. For both datasets, we use random horizontal flip, random crops to augment the data on the fly and use We used $\alpha = 0.3$ for the importance memory update and $K = 4$ for the pruning factor.

For CIFAR-10, we also trained a Vision Transformer (ViT) (Dosovitskiy et al., 2021) using the Adam optimizer with an initial learning rate 0.0001, divided by 10 after 70, 140 epochs. Here we also use random horizontal flip, random crops to augment the data on the fly and use $\alpha = 0.3$ for the importance memory update and $K = 8$ for the pruning factor.

**Point cloud classification.** We train a PointNet (Qi et al., 2017) with 3 shared-MLP layers and one fully-connected layer, on the ModelNet40 dataset (Wu et al., 2015). The dataset contains point clouds from 40 categories. The data are split into 9,843 for training and 2,468 for testing. Each point cloud has 1,024 points. We use the Adam optimizer with batch size 64, weight decay 0.001, initial learning rate 0.00002 divided by 10 after 100, 200 epochs. We train for 300 epochs in total We used $\alpha = 0.0$ and $K = 8$ for our methods.

**Oxford Flower-102.** The Oxford 102 flower dataset (Nilsback & Zisserman, 2008) contains flower images from 102 categories. We follow the same experiment setting of Zhang et al. (2017; 2019). We use the original test set for training (6,149 images) and the original training set for testing (1,020 images). In terms of network architecture, we use the pre-trained VGG-16 network (Simonyan & Zisserman, 2014) for feature extraction and only train a two-layer fully-connected network from scratch for classification. We use the Adam optimizer with a learning rate 0.001 and train the two-layer fully-connected network for 100 epochs. We used $\alpha = 0.2$ and $K = 8$ for our methods.

**Tiny-ImageNet.** Tiny-ImageNet dataset is proposed by Le & Yang (2015). Tiny-ImageNet is a larger dataset contains 100,000 training examples from 200 categories. We train a ResNet-18 network (He et al., 2016) for 20 epochs with a batch size of 64, a learning rate of 0.001 divided by 10 after 10 epochs, SGD optimizer with momentum 0.9 and data augmentation of random horizontal flip. We used $\alpha = 0.3$ and $K = 64$ for our methods.

**Image regression.** The image regression task involves training a network to learn a 2D image signal, where each pixel is treated as an individual data point. The input to the network is the pixel's 2D coordinates, and the output is the corresponding RGB value for that pixel. We trained a 5-layer SIREN network (Sitzmann et al., 2020) for 300 epoch using sinusoidal encoding for the

input coordinates, optimized with Adam at a learning rate of 0.0003 and a batch size of 512. For our method we used $\alpha = 0.3$ and $K = 2$.

## D  ADDITIONAL EXPERIMENTS

In this section we present additional experiments.

Table 2 presents the computation times for four different importance metrics used in Fig. 2. Our importance metric is nearly as fast as simply using the loss for classification tasks, thanks to its analytic form. When utilizing automatic differentiation, the computation time of our metric is comparable to DLIS, as both require backpropagation on the network output (for ours) or the last layer (for DLIS). The final method, SRGD, involves a significantly more costly metric evaluation.

Table 3 presents a comparison between various methods, including Uniform sampling, Loss-based importance sampling, DLIS, DLIS with our algorithm, LOW, our method using only importance sampling, and a combination of importance sampling with data pruning across multiple tasks. For each task, we report the cross-entropy loss and accuracy at both equal steps and equal time, alongside the total optimization time for each method. Overall, our method, which combines importance sampling and data pruning, achieves the best results in terms of both equal time and equal steps. Although it does not always yield the top result, it frequently ranks second or first. Some specific cases are worth highlighting. In the CIFAR-10 classification task using a ViT network, we observe longer training times for our method, even with data pruning. This is due to the overhead introduced by gradient backpropagation and the relatively low level of pruning. This overhead is also noticeable in both DLIS methods. In the image regression task, where each pixel is treated as a separate data point, the memory footprint is low. Consequently, data pruning does not lead to sufficient time reduction to offset the overhead introduced by importance sampling and its computation, which results in Uniform sampling outperforming our method in terms of computation time. Additionally, there are instances where our method using only importance sampling outperforms the version with data pruning. This occurs in cases where significant data pruning negatively impacts generalization. For instance, in the Flower-102 dataset, each class contains only a few examples, so pruning too many data points directly compromises the model's training capacity, as each point carries critical information.

Finally, we present three convergence curves—showing both loss and accuracy—under equal time and equal steps for CIFAR-10 (ViT) (Fig. 5), Point Cloud (Fig. 6), and Tiny-ImageNet (Fig. 7) classification tasks. These curves illustrate the evolution of classification error across the methods used in Table 3. Our method consistently achieves superior performance throughout the training process. Additionally, the results highlight significant underperformance for certain methods, such as DLIS on Tiny-ImageNet and CIFAR-10 (ViT). The results also emphasize the substantial difference between equal steps and equal time for the LOW method, which suffers from a considerable overhead. Although LOW performs well under equal steps, it can perform worse than Uniform sampling when evaluated at equal time.

Table 2: Average computation time on 3 layer fully-connected network for multiple sampling metric the task from Fig. 2. Time is average over one epoch and computed on mini-batch of size 8.

| Computation time ($\downarrow$) | Loss | SRGD | DLIS | Ours autodiff | Ours analytic |
|---|---|---|---|---|---|
| Polynomial regression | $1.33 \cdot 10^{-4}$ $(1. \times)$ | $7.17 \cdot 10^{-4}$ $(5.39 \times)$ | $4.38 \cdot 10^{-4}$ $(3.29 \times)$ | $3.23 \cdot 10^{-4}$ $(2.43 \times)$ | - |
| MNIST | $1.28 \cdot 10^{-4}$ $(1. \times)$ | $5.57 \cdot 10^{-4}$ $(4.35 \times)$ | $3.27 \cdot 10^{-4}$ $(2.55 \times)$ | $3.59 \cdot 10^{-4}$ $(2.80 \times)$ | $1.40 \cdot 10^{-4}$ $(1.09 \times)$ |

Table 3: We compare the impact of importance sampling (IS) with and without data pruning on classification and regression tasks. Our approach consistently outperforms on majority of datasets. Bold numbers represents the best scores, underlined ones represent the second best. Worse to best for each metric is shown from red to green.

| Dataset | Method | Equal step | | | Equal time | |
|---------|--------|------------|--|--|------------|--|
| | | Loss (↓) | Accuracy (↑) | Time(s)(↓) | Loss | Accuracy |
| MNIST | Uniform | 0.00092 | 97.5 | 477 | 0.00097 | 97.4 |
| | Loss IS | 0.00083 | 97.8 | 474 | 0.00086 | 97.7 |
| | DLIS | 0.00106 | 98.0 | 754 | 0.00124 | 97.7 |
| | DLIS weights **w/ Our algorithm** | 0.00083 | 97.8 | 665 | 0.00089 | 97.6 |
| | LOW | 0.00072 | 98.1 | 624 | 0.00077 | 98.0 |
| | **Our IS** | 0.00083 | 97.8 | 395 | 0.00085 | 97.7 |
| | **Our IS + Data pruning** | **0.00056** | **98.3** | **386** | **0.00059** | **98.2** |
| CIFAR-10 | Uniform | 0.0037 | 92.5 | 5810 | 0.0037 | 92.4 |
| | Loss IS | 0.0060 | 84.6 | 5797 | 0.0057 | 82.6 |
| | DLIS | 0.0055 | 84.6 | 5963 | 0.0036 | 89.1 |
| | DLIS weights **w/ Our algorithm** | 0.0105 | 63.1 | 5934 | 0.0127 | 53.7 |
| | LOW | 0.0036 | 92.5 | 10768 | 0.0039 | 91.0 |
| | **Our IS** | 0.0038 | 92.6 | 5836 | 0.0036 | 92.4 |
| | **Our IS + Data pruning** | **0.0034** | **92.8** | **4395** | **0.0034** | **92.8** |
| CIFAR-10 (ViT) | Uniform | 0.00794 | 74.6 | 7105 | 0.00789 | 74.8 |
| | Loss IS | 0.00790 | 74.7 | **7039** | 0.00779 | 75.1 |
| | DLIS | 0.00932 | 67.6 | 9111 | 0.00917 | 68.2 |
| | DLIS weights **w/ Our algorithm** | 0.01050 | 63.6 | 8916 | 0.01000 | 65.1 |
| | LOW | **0.00762** | 74.6 | 13046 | 0.00788 | 73.7 |
| | **Our IS** | 0.00785 | **75.5** | 7113 | **0.00769** | **75.7** |
| | **Our IS + Data pruning** | 0.00786 | 75.2 | 7213 | 0.00785 | 75.3 |
| CIFAR-100 | Uniform | 0.012 | 72.6 | 4748 | 0.012 | 73.8 |
| | Loss IS | 0.023 | 40.2 | 4404 | 0.026 | 0.02632.8 |
| | DLIS | 0.015 | 62.0 | 4501 | 0.015 | 60.8 |
| | DLIS weights **w/ Our algorithm** | 0.0260.021 | 0.026 43.6 | 4603 | 0.029 | 26.3 |
| | LOW | 0.011 | **74.6** | 4730 | 0.011 | 74.2 |
| | **Our IS** | 0.011 | 74.3 | 4686 | 0.012 | 74.3 |
| | **Our IS + Data pruning** | **0.011** | 74.3 | **3150** | **0.011** | **74.3** |
| Flower-102 | Uniform | **0.00658** | **79.8** | 6327 | **0.00658** | **79.8** |
| | Loss IS | 0.01777 | 57.4 | 5909 | 0.01777 | 57.4 |
| | DLIS | 0.02128 | 46.9 | 7399 | 0.01951 | 43.8 |
| | DLIS weights **w/ Our algorithm** | 0.72899 | 30.9 | 7504 | 0.33971 | 27.8 |
| | LOW | 0.00773 | 76.1 | 8241 | 0.00755 | 76.7 |
| | **Our IS** | 0.00689 | 79.5 | 6263 | 0.00689 | 79.5 |
| | Ours IS + Data pruning | 0.01353 | 73.6 | **3356** | 0.01353 | 73.6 |
| Point cloud | Uniform | 0.00505 | 82.3 | 356 | 0.00505 | 82.2 |
| | Loss IS | 0.00495 | 82.6 | 358 | 0.00496 | 82.5 |
| | DLIS | 0.00595 | 81.9 | 580 | 0.00603 | 81.8 |
| | DLIS weights **w/ Our algorithm** | 0.00481 | 82.6 | 561 | 0.00485 | 82.5 |
| | LOW | 0.00572 | 82.6 | 4714 | 0.01173 | 74.9 |
| | **Our IS** | 0.00480 | 82.9 | 374 | 0.00480 | 82.9 |
| | **Our IS + Data pruning** | **0.00478** | **83.2** | **354** | **0.00478** | **83.1** |
| Tiny-ImageNet | Uniform | 0.02176 | 47.4 | 7602 | 0.02176 | 47.4 |
| | Loss IS | 0.02237 | 45.2 | 8407 | 0.02238 | 45.2 |
| | DLIS | 0.03433 | 26.3 | 7897 | 0.03433 | 26.3 |
| | DLIS weights **w/ Our algorithm** | 0.03454 | 26.1 | 9300 | 0.02778 | 35.0 |
| | LOW | 0.02344 | 43.5 | 7702 | 0.02344 | 43.5 |
| | **Our IS** | **0.02127** | **48.1** | 8461 | **0.02123** | **48.5** |
| | **Our IS + Data pruning** | 0.02193 | 47.1 | **4349** | 0.02193 | 47.1 |
| Image regression | Uniform | 9.44 | - | **2308** | 9.44 | - |
| | Loss IS | 11.01 | - | 2360 | 11.12 | - |
| | DLIS | 14.27 | - | 2949 | 15.78 | - |
| | DLIS weights **w/ Our algorithm** | 8.44 | - | 2863 | 9.37 | - |
| | **Our IS** | 8.13 | - | 2912 | 9.16 | - |
| | **Our IS + Data pruning** | **8.01** | - | 2328 | **8.05** | - |

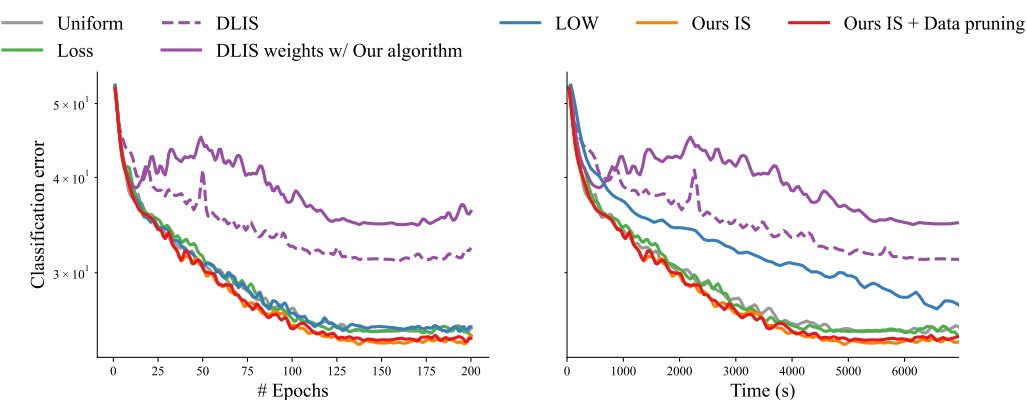

Figure 5: Comparisons on CIFAR-10 using Vision Transformer (ViT) (Dosovitskiy et al., 2021). The results show consistent improvement of Ours IS and Ours IS + Data pruning over LOW (Santiago et al., 2021) and DLIS (Katharopoulos & Fleuret, 2018) for both equal epoch and equal time.

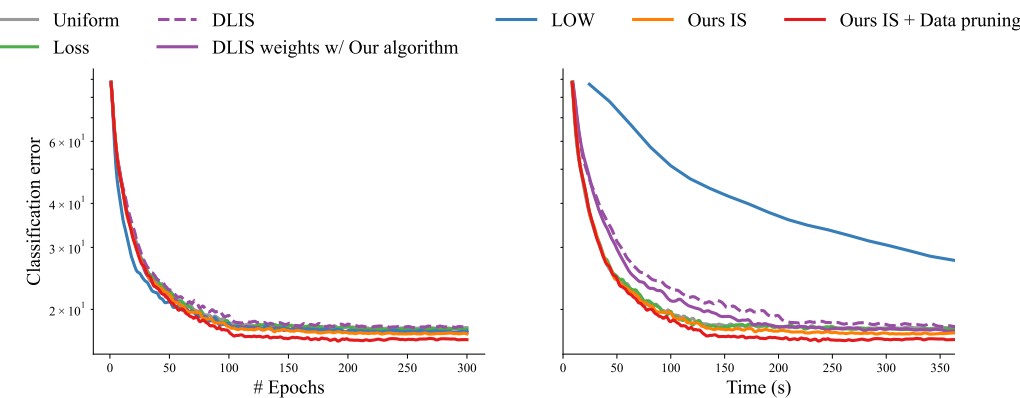

Figure 6: When comparing on the Point cloud (ModelNet40 (Wu et al., 2015)) classification dataset, DLIS performs poorly at equal time due to the resampling overhead. Unlike DLIS (Katharopoulos & Fleuret, 2018), we use standard uniform sampling which is faster. We also compare against another adaptive scheme by Santiago et al. (2021) (LOW). Our importance sampling (Ours IS) with data pruning (Ours IS + Data pruning) show improvements on the ModelNet40 dataset against other methods. achieving lower classification errors with minimal overhead compared to others.

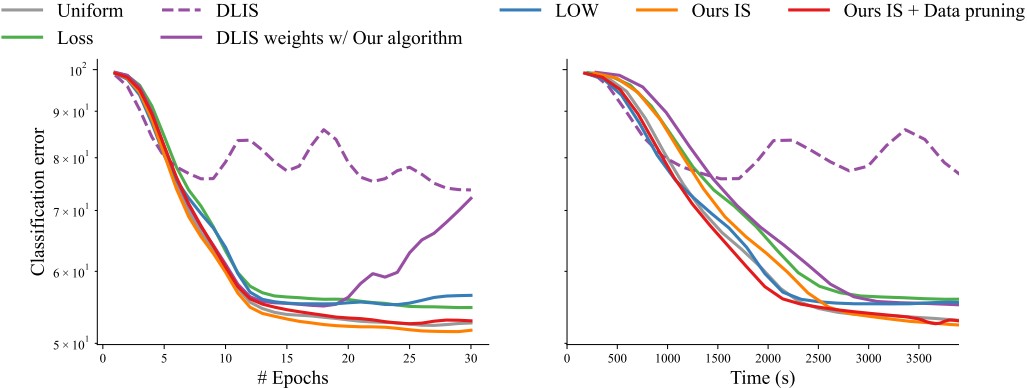

Figure 7: Comparisons on Tiny-ImageNet (Le & Yang, 2015). The results show improvement of our importance sampling (Ours IS) over other methods, while in this case Ours IS with data pruning works similarly to Uniform sampling.