# OpenReview forum: "Online importance sampling for stochastic gradient optimization"
_ICLR.cc/2025/Conference — ICLR 2025 Conference Withdrawn Submission_

### Official Review · Reviewer_UAi1 · 2024-10-15

**Soundness:** 3
**Presentation:** 2
**Contribution:** 2
**Rating:** 5
**Confidence:** 4

**Summary:**

This work studies stochastic gradient descent with online importance sampling and data pruning. The authors propose a practical metric that requires little computation and use it both for the importance weights and pruning scores. This metric is updated on the fly during training. The authors then evaluate their framework on multiple tasks such as classification and regression and popular benchmark datasets such as Cifar10/100 and MNIST.

**Strengths:**

The authors evaluated their proposed framework on multiple datasets and different tasks. The results look promising and show some gains compared to the competitors that seem consistent across tasks.

**Weaknesses:**

1) I found the writing in Sections 3 and 4 to be unclear and somewhat lacking in precision; they would need rewriting and clarifications in my opinion.
For example, equation (1) is confusing, as it seems different from the typical loss minimized, which is generally expressed as $\mathbb{E}(\mathcal{L}(m(x, \theta), y))$, where $x,y$ follow a given data generating process. I fail to understand this renormalization and what $p(x,y)$ refers to here: is it the data-generating process or importance sampling weights? Could you explain and define $p(x,y)$.
Besides, there are some assumptions that are not clearly stated and appear here and there in the text, for example, that the model is Lipschitz (Section 4.1).

2) The proposed metric seems to be the same as the EL2N proposed in Deep Learning on a Data Diet, Paul et al. Could the authors explain the difference?

3) Although the proposed method seems to outperform the other methods consistently, the differences are sometimes very small, and it is difficult to know if we can not attribute it to statistical noise. The authors indicate that they average over 3 runs; it would be interesting to quantify the variability of the results.

**Questions:**

See above

---

### Official Review · Reviewer_VTtH · 2024-11-03

**Soundness:** 2
**Presentation:** 3
**Contribution:** 1
**Rating:** 3
**Confidence:** 4

**Summary:**

The authors propose to use the derivative wrt to the network output for importance sampling. They also propose that their method can be used for online data pruning. They demonstrate their performance is stronger than some of the baselines.

**Strengths:**

1. Overall, the writing is clear and easy to follow.

**Weaknesses:**

There are many papers in the literature regarding importance sampling and the authors seem not to know many of them. It is true that the methods proposed in the earlier years have the problem of computation efficiency, but nowadays there are many methods with little overhead.


1. The proposed method has limited novelty. While the authors claim that the derivative wrt to the network output is different from what is used in (Katharopoulos&Fleuret,2018), from my understanding, it is pretty similar if not identical, and there are many other works that utilize something similar like the EL2N score in [1]. Even if there is some difference, the novelty seems limited and the changes are not argued or justified. (Like the difference may just be taking the derivative to the pre- or post- activation output)
2. The experiment comparison is weak. They do not compare with some more recent work like [2]. Additionally, they do not compare with some simple and already used baseline in practice. For example, random shuffling with a reduced number of epochs is often stronger than many of these importance sampling methods. (This is sometimes much stronger than uniform sampling as the samples won’t repeat within an epoch. Also, it is important that the learning rate schedule changes so that the learning rate decays)



[1] Paul, M., Ganguli, S., & Dziugaite, G. K. (2021). Deep learning on a data diet: Finding important examples early in training. Advances in neural information processing systems, 34, 20596-20607.

[2] Qin, Z., Wang, K., Zheng, Z., Gu, J., Peng, X., Zhou, D., ... & You, Y. InfoBatch: Lossless Training Speed Up by Unbiased Dynamic Data Pruning. In The Twelfth International Conference on Learning Representations.

**Questions:**

1. Can you clarify the novelty of your proposed metric and why should it be better than those used in the literature?
2. Can you add more comparisons in the experiments with more recent works ([2] and a lot more) and also the most simple random reshuffling with a reduced number of epochs?
3. Can you add more details regarding sampling with or without replacement? It is said within a batch, it is sampled without replacement, so the samples will not repeat. But what about for different batches from the same epoch. To me, it seems like samples can repeat within an epoch, and this can lead to inferior performance compared to those will not repeat within an epoch such as [2].

---

### Official Review · Reviewer_usHw · 2024-11-04

**Soundness:** 1
**Presentation:** 2
**Contribution:** 1
**Rating:** 3
**Confidence:** 3

**Summary:**

This paper proposes a new adaptive method of importance sampling for stochastic gradient estimation in multi-class classification. The sampling weights of this method do not depend on the costly full backpropagation on each data point. The importance sampling weights can also be used for data pruning, which can be further combined with the importance sampling for gradient estimation. The authors conducted experiments on classification tasks to verify the effectiveness of their algorithm compared to SGD with uniform sampling and previous importance sampling methods.

**Strengths:**

- The idea of using importance sampling weights for data pruning is interesting.
- The plots in Figure 1 numerically verify that the learned importance sampling weights are somewhat meaningful and provide intuitions of why this method could improve upon uniform sampling.

**Weaknesses:**

- A major concern with the importance sampling method proposed in this paper is that it remains "loss-based". To be specific, the weight of each data $x$ is proportional to $\\|\frac{\partial \mathcal{L}(x)}{\partial m(x,\theta)}\\|\_2 = \sqrt{\sum_{j=1}^J(s\_j(x) - y\_j(x))^2}$, where $s\_j(x)$ is the predicted probability of data $x$ belongs to class $j$ while $y\_j(x)$ is the groundtruth. Thus, the importance sampling weight of data $x$ can be viewed as its $\ell_2$ loss on label prediction. However, it is unclear how this approach relates to the theoretically optimal importance sampling weight based on gradient norms. If the gradient w.r.t. the output does not take the specific form as in the logistic loss, does it still make sense to sample based on the norm of the gradient w.r.t the output?
- There is no formal convergence analysis of the proposed algorithm. So the algorithm remains heuristic.
- The experiments in this paper were not repeated with different random seeds, resulting in a lack of error bars on reported values and curves. This makes their experimental results less reliable.

**Questions:**

- Why (6) is only for binary classification tasks (as indicated in Line 205)? Could $J$ be larger than 2?
- In [1], the authors mention that ``The individual parameter derivatives vary uniquely across the data points, and estimation using a single distribution inevitably requires making a trade-off" and advocate for the multiple importance sampling (MIS) approach. Could you please comment on this? Could you experimentally compare their MIS-based algorithm with your algorithm?

[1] Salaün, Corentin, Xingchang Huang, Iliyan Georgiev, Niloy J. Mitra, and Gurprit Singh. "Multiple importance sampling for stochastic gradient estimation." arXiv preprint arXiv:2407.15525 (2024).

**Details Of Ethics Concerns:**

This paper has substantial overlaps with the paper [1] on arXiv. For example, Section 3 in this paper is almost the same as Section 3 in [1], Eq. 5 in this paper is the same as (4) in [1], The figure in line 270 is the same as the one in Page 4 of [1], Algorithm 1 in this paper is the same as Algorithm in [1], and so on.

Although this paper might be a resubmission on top of [1] by the same authors, it is still quite weird that [1] is not properly cited and compared as a related work since the experimental results in this paper are quite different from those in [1] and the main selling points of these two papers are intrinsically different: [1] highlights the multiple importance sampling method while this paper is purely based on importance sampling with a single distribution.

[1] Salaün, Corentin, Xingchang Huang, Iliyan Georgiev, Niloy J. Mitra, and Gurprit Singh. "Multiple importance sampling for stochastic gradient estimation." arXiv preprint arXiv:2407.15525 (2024).

---

### Note · Authors · 2024-11-19

I have read and agree with the venue's withdrawal policy on behalf of myself and my co-authors.